# A_2_-Mode Lamb Passive-Wireless Surface-Acoustic-Wave Micro-Pressure Sensor Based on Cantilever Beam Structure

**DOI:** 10.3390/s25061873

**Published:** 2025-03-18

**Authors:** Zhuoyue Duan, Tao Wang, Wei Ji, Lihui Feng, Peng Yin, Jihua Lu, Litong Yin

**Affiliations:** 1School of Information Science and Engineering, Shandong University, Qingdao 266237, China; 201700121041@mail.sdu.edu.cn (Z.D.); 202232716@mail.sdu.edu.cn (T.W.); 2School of Optoelectronic, Beijing Institute of Technology, Beijing 100081, China; lihui.feng@bit.edu.cn; 3School of Cyberspace Security, University of Chinese Academy of Sciences, Beijing 100049, China; yinpeng@iie.ac.cn; 4School of Integrated Circuits and Electronics, Beijing Institute of Technology, Beijing 100081, China; lujihua@bit.edu.cn (J.L.); 3220221544@bit.edu.cn (L.Y.)

**Keywords:** SAW micro-pressure sensor, passive wireless, cantilever beam, MEMS-compatible manufacturing process

## Abstract

Passive-wireless surface-acoustic-wave (SAW) micro-pressure sensors are suitable for extreme scenarios where wired sensors are not applicable. However, as the measured pressure decreases, conventional SAW micro-pressure sensors struggle to meet expected performance due to insufficient sensitivity. This article proposes a a method of using an $A_{2}$-mode Lamb SAW sensor and introduces an inertial structure in the form of a cantilever beam to enhance sensitivity. An MEMS-compatible manufacturing process was employed to create a multi-layer structure of ${\mathrm{SiO}}_{2}$, AlN, and SOI for the SAW micro-pressure sensor. To investigate the operational performance of the SAW micro-pressure sensor, a micro-pressure testing system was established. The experimental results demonstrate that the sensor exhibits high sensitivity to micro-pressure, validating the effectiveness of the proposed design.

## 1. Introduction

In recent years, increasing demand in fields such as biomedicine, aerospace, and environmental monitoring has spurred advances in micro-pressure measurement, leading to the development of diverse sensor technologies [1,2]. Traditional wired sensors, while widely used, face inherent limitations: their physical connections complicate installation and restrict deployment in harsh or inaccessible environments [3]. Wireless sensors, by contrast, eliminate dependency on cumbersome cabling, enhancing operational flexibility and simplifying deployment. These wireless systems are further categorized by power source: active sensors rely on internal batteries, which necessitate periodic maintenance and struggle under extreme conditions (e.g., high temperatures, corrosive atmospheres), while passive sensors circumvent these issues by harvesting energy directly from ambient electromagnetic signals. This energy autonomy enables passive sensors to operate maintenance-free over extended periods, even in demanding settings. Combining the benefits of cable-free design and battery independence, SAW wireless passive sensors have emerged as a robust solution for precision micro-pressure monitoring in challenging applications [4]. This dual capability makes them necessary for applications where long-term, maintenance-free operation in extreme or hard-to-access environments is critical, such as in implantable biomedical devices, aircraft engine monitoring, or underground environmental sensing.

SAW micro-pressure sensors are widely used due to their passive wireless operation, straightforward and intuitive sensing signal variation process, and compact size. For instance, Li et al. designed a SAW micro-pressure sensor based on a piezoelectric film of ${\mathrm{LiNbO}}_{3}$ using finite element analysis in 2015 [5]. In 2021, Y Li and colleagues proposed a shear horizontal SAW micro-pressure sensor with a grooved structure [6]. However, as the accuracy of the measured parameters continues to improve, SAW pressure sensors face challenges in achieving the desired sensitivity. In recent years, numerous studies have suggested enhancing sensor sensitivity by introducing inertial structures such as cantilever beams. For example, in 2021, Mo et al. designed a cantilever beam pressure sensor based on an optical micro-ring resonator, which achieved an eightfold increase in sensitivity compared to sensors with non-cantilever beam structures [7]. In 2022, Z. Xin and colleagues designed a high-performance dual-channel MEMS microwave power sensor featuring a cantilever beam, achieving a sensitivity of 82.9 mV/W for the thermoelectric detection channel [8]. Wu et al. introduced a cantilever beam structure made of silicon in a gas sensor designed in 2022 for high-sensitivity methane detection, achieving a sensitivity of 2.0 pm/ppm [9]. To highlight the differences, Table 1 compares the key parameters of SAW pressure sensors across various design configurations.

From the above literature, it is evident that the introduction of inertial structure cantilever beams has significantly enhanced the sensitivity of SAW micro-pressure sensors. Based on these findings, this study designed a passive wireless $A_{2}$-mode Lamb micro-pressure sensor based on a cantilever beam structure, using Lamb’s high-mode $A_{2}$ sensor [14,15]. Compared to the fundamental Lamb mode, this design exhibits enhanced sensitivity, faster responses, and less loss, achieving a Q factor of up to 442. The introduction of the inertial structure cantilever beam improves the sensitivity from 500 Hz/Pa to 25 kHz/Pa, resulting in a fifty-fold increase in sensitivity.

The first part of this article introduces the problems and shortcomings of surface-acoustic-wave micro-pressure sensors, as well as the benefits of introducing cantilever beam structures. The second part introduces the design and parameter optimization of SAW micro-pressure sensor system. The third part introduces the MEMS process preparation of the SAW micro-pressure sensors. The fourth part introduces performance analysis and system testing. The fifth part is the conclusion.

## 2. System Design and Parameter Optimization

### 2.1. Design of Micro-Pressure Sensing System

Figure 1 illustrates the schematic diagram of the SAW micro-pressure sensor system, which includes a radio frequency transceiver system and the SAW micro-pressure sensor. The sensor comprises an Interdigital Transducer (IDT), a piezoelectric film of aluminum nitride (AlN), and a silicon substrate (Si) [16]. The piezoelectric film and the substrate together form a cantilever beam structure.

The IDT of the designed SAW micro-pressure sensor employs a dual-port resonant structure, which includes an input IDT, an output IDT, an Open Circuit Reflective Grating (OCRG), and a Short Circuit Reflective Grating (SCRG) [17]. During operation, the RF transceiver emits a specific frequency RF signal through an antenna to excite the SAW micro-pressure sensor. After the sensor receives this signal via the antenna, it passes through a circulator to reach the input IDT, where the inverse piezoelectric effect converts the RF signal into surface acoustic waves (SAW) that propagate laterally along the cantilever beam. The input and output IDTs, along with the left and right reflective gratings, form a resonant cavity that not only reduces losses but also enhances frequency selection characteristics. The output IDT converts the SAW back into an RF signal under the piezoelectric effect for transmission. The RF transceiver receives and demodulates this RF signal, allowing the target physical quantity to be determined through the relationship between pressure and frequency [18].

The resonant frequency $f_{0}$ of the SAW sensor in operational conditions is governed by the fundamental relationship(1)$$f_{0}=\frac{v}{2p}$$
where *p* represents the interdigital transducer (IDT) electrode pitch and *v* denotes the SAW propagation velocity. Through differential analysis of Equation (Equation 1), we derive the frequency shift expression as(2)$$\Delta f_{r}=\frac{1}{2}\left(\frac{\Delta v}{p}-\frac{\Delta p\cdot v}{p^{2}}\right)$$

Here, Δv and Δp correspond to the variations in SAW velocity and IDT electrode pitch, respectively, induced by mechanical stress when the piezoelectric cantilever beam experiences external loading.

The operational principle of the micro-pressure sensor can be elucidated through cantilever beam stress analysis. This analysis reveals that the resonant frequency shift $\Delta f_{r}$ exhibits a linear dependence on the applied force *F* at the free end, expressed as(3)$$\Delta f_{r}=\frac{6Fl}{Ewh^{2}}\left[\left(r-1\right)-\mu r_{2}\right)]f_{0}$$
where *l*, *w*, and *h* respectively denote the cantilever’s length, width, and thickness. The fundamental resonant frequency $f_{0}$ serves as the baseline parameter, while material-dependent constants include Young’s modulus *E*, Poisson’s ratio μ, and dimensionless coefficients $r_{1}$ and $r_{2}$.

This derived linear relationship between the SAW sensor’s frequency shift $\Delta f_{r}$ and the applied pressure *F* establishes the foundation for precise pressure measurement. The proportionality constant incorporates both geometric parameters of the cantilever and intrinsic material properties, enabling sensor optimization through dimensional control and material selection.

### 2.2. Parameter Optimization of SAW Micro-Pressure Sensors

#### 2.2.1. Piezoelectric Films

Piezoelectric films serve as crucial mediums for exciting SAW signals, fulfilling two primary functions in SAW micropressure sensors: first, they act as sensitive elements that detect minute changes in external pressure; second, they function as the excitation and propagation medium for SAW signals, facilitating the conversion between radio frequency signals and SAW through the direct and inverse piezoelectric effects. Common piezoelectric film materials include quartz, lithium niobate, and aluminum nitride. The piezoelectric materials must meet the following criteria: (1) high SAW wave velocity; (2) low propagation loss and stable transmission; (3) ease of processing. A comparison of the parameters of several common piezoelectric film materials is presented in Table 2. Among these, aluminum nitride demonstrates significant advantages in SAW wave velocity and electromechanical coupling coefficient compared to other materials, leading to the final selection of aluminum nitride as the piezoelectric substrate material for this experiment.

#### 2.2.2. Cantilever Beam

The cantilever beam can amplify stress and enhance the sensitivity of sensor detection. The stress distribution at different positions after applying pressure to the cantilever beam is illustrated in Figure 2a. As shown in the figure, the left end of the beam was fixed, while the right end was free. The red area indicates the region of maximum stress, particularly near the fixed end, where the stress reached its peak value. The IDT could be positioned close to the fixed end of the cantilever beam. Due to process limitations and the performance of the IDT, the thickness of the cantilever beam was determined to be 15.6 μm, which included a 0.6 μm thick piezoelectric film (AlN) and a 15 μm thick silicon substrate. To investigate the impact of different lengths of the cantilever beam on amplification capability, the deformation displacement at the free end under different lengths was compared, with the lengths ranging from 4000 to 8000 μm, and both width and thickness set at 1000 μm and 15.6 μm, respectively. The applied pressure varied from 0.1 to 0.19 Pa, and the results are presented in Figure 2b. It was concluded that a longer cantilever beam produced greater displacement under the same pressure, and the slope of the displacement change with respect to pressure was steeper, resulting in higher sensitivity. In the absence of external forces, considering only the effect of gravity, the stress near the fixed end could be calculated using Equation (Equation 4).(4)$$\sigma =\frac{wgL^{2}c}{8I}$$

In this context, w represents the mass of the cantilever beam, *g* denotes the acceleration due to gravity, *L* indicates the length of the cross-section of the cantilever beam, *c* is the distance from the fixed end to the neutral axis of the cross-section, and *I* refers to the moment of inertia of the cross-section. By substituting specific values into Equation (Equation 4), the results are depicted in Figure 2c, showing that the bending strength of AlN was 350 MPa [19]. When the length of the cantilever beam reached 8500 μm, it would fracture and cease to function properly. Considering the measurement range of the sensor, a cantilever beam structure with a length of 7000 μm was selected.

#### 2.2.3. Integrated Digital Terminal

(1) Forked cycle *p*

In order to match the RF transceiver system, the resonant frequency of the SAW sensor needed to be within the range of 420–450 MHz. The thickness of the cantilever beam was 15.6 μm. Figure 3a shows the dispersion pattern of Lamb in an aluminum plate. As the product of the resonant frequency and the thickness of the cantilever beam gradually increased, higher-order Lamb continuously appeared and the velocity gradually decreased towards stability. The different modal shapes of Lamb in AlN and Si were obtained through simulation calculations, as shown in Figure 3c. The resonant frequencies of asymmetric modes $A_{0}$, $A_{1}$, and $A_{2}$ were 193 MHz, 283 MHz, and 438 MHz, respectively. The resonant frequencies of symmetric modes $S_{0}$, $S_{1}$, and $S_{2}$ were 232 MHz, 330 MHz, and 467 MHz, respectively. Within this range, in order to ensure a high SAW wave velocity and comply with the frequency range, $A_{2}$-order Lamb was selected, and the calculated resonant frequency was 438 MHz. At this time, the Lamb wave velocity was 14,892 m/s and the forked period was 34 μm. The width and spacing of the fingers were both 8.5 μm.

(2) Forked cycle

The cross-point logarithmic *N* affects the bandwidth of a device; the smaller the bandwidth, the higher the frequency selection sensitivity. The expression is as follows:(5)$$BW=\frac{2f}{N}$$

As the number of interdigital finger pairs *N* increases, the selectivity of the filter improves. However, an excessive number of finger pairs can lead to increased insertion loss and greater difficulty in fabrication. In this study, under the conditions permitted by the fabrication process, the chosen number of interdigital finger pairs was 20.

(3) Electrode height *H*

After depositing IDT electrodes on the piezoelectric thin film, the mass loading effect of the electrodes reduced the SAW wave velocity. To quantitatively analyze the secondary effects of electrode parameters, we established a series of finite element models with varying electrode heights *H* and conducted modal analysis on them. We extracted the characteristic frequency of each model and converted it into SAW wave velocity. However, the electrode height could not be too low as this would severely deteriorate the conductivity of the IDT. This study simulated the relationship between electrode height in the range of 0.2–0.6 μm and SAW wave velocity, as shown in Figure 3b. With the increase in electrode height *H*, the resonant frequency of SAW decreased and, correspondingly, the velocity of the surface wave also decreased. To maximize the SAW wave velocity while ensuring stable excitation, the optimal height for the IDT electrodes was 0.2 μm, at which the SAW wave velocity was 14,802 m/s.

#### 2.2.4. Reflective Grating Design

Reflective gratings are key structures in resonant SAW devices and reflect the SAW emitted from IDTs back to the IDTs, thus forming a resonant cavity. A reflective grating consists of periodically arranged metal electrodes. The SAW excited by the IDT reflects back and forth between the reflective gratings on both sides. Within certain specific frequency bands, the in-phase SAWs superimpose on each other, allowing for the filtering of SAWs at different frequencies. Common types of reflective gratings include open-circuit and short-circuit gratings. When SAW encounters the former, reflection occurs without a phase change, whereas the latter induces both reflection and a phase change. The distance between the reflective grating and the IDT affects the resonant cavity, and an appropriate distance can create strong standing waves within the cavity [20]. When the distance between the OCRG (open-circuit reflective grating) and the IDT satisfies Equation (Equation 6), the phase difference is an integer multiple of 2π:(6)$$S=\frac{\lambda }{2}n+\frac{\lambda }{4}$$

The wavelength of SAW is denoted by λ and *n* is a positive integer. The distance between SCRG and IDT satisfies Equation (Equation 7):(7)$$S=\frac{\lambda }{2}n$$

The period of the diffraction grating must satisfy the condition for Bragg reflection, as shown in Equation (Equation 8) [21]:(8)2p=nλ

The period of the reflective grating is denoted by *p* and *n* is a positive integer. Substituting the parameters into the above formula and setting *n* to 1, the calculated distances of OCRG and SCRG from IDT were 17 μm and 8.5 μm, respectively, with both the electrode width and period being 8.5 μm. Figure 4 explores the simulations of various configurations: without reflective gratings, with OCRG at both ends, with SCRG at both ends, and a combination of OCRG and SCRG. Comparing these simulation results, it is evident that the introduction of reflective gratings resulted in lower losses than configurations without them. The combination of OCRG and SCRG yielded the best performance, exhibiting more pronounced filtering characteristics and lower losses compared to using OCRG or SCRG at both ends.

Figure 5 represents the model after parameter optimization and Table 3 details the model’s parameters. The top layer consisted of 50 OCRGs, 20 pairs of input IDTs, 20 pairs of output IDTs, and 50 SCRGs. The spacing and width of these components were uniformly 8.5 μm, with an electrode height of 200 nm. The cantilever beam was composed of a 600 nm thick piezoelectric AlN film on top, followed by a 15 μm thick silicon layer beneath.

## 3. Preparation of SAW Micro-Pressure Sensor

### 3.1. Layout Determination

Figure 6 presents the layout design of the SAW micro-pressure sensor discussed in this paper. During the actual fabrication process, four photolithography steps were required: The first step involved the patterning of the input–output IDT, reflective grating, and connection pad, which linked the sensor’s input and output IDTs to the SMA leads. The second step focused on the pad pattern; after the electrode fabrication, the sputtered ${\mathrm{SiO}}_{2}$ protective layer covered the connection pads, necessitating photolithography to etch the pad pattern for exposure. The third step involved the front patterning of the cantilever beam. At this stage, the materials from top to bottom were ${\mathrm{SiO}}_{2}$, AlN, Si, ${\mathrm{SiO}}_{2}$, and Si. A single etching step could not fully release the cantilever beam, so photolithography was used to pattern the three edges of the cantilever beam. The fourth step pertained to the back cavity pattern of the cantilever beam.

### 3.2. Device Manufacturing

Based on the theoretical results from the previous chapter, we designed a SAW micro-pressure sensor with a resonant frequency of 438 MHz using a silicon-on-insulator (SOI) wafer with a top silicon thickness of 15 μm. Figure 7 illustrates the fabrication process of the device. (a) The surface of the SOI wafer was cleaned to prevent contamination during the AlN sputtering process. (b) A 600 nm piezoelectric AlN film was deposited. (c) Negative photoresist was applied for photolithography. (d) The mask was exposed to ultraviolet light. (e) During development, the unexposed areas of the negative resist were removed. (f) A 200 nm aluminum electrode was sputtered, and the electrode adhering to the photoresist was stripped using NMP solution. After stripping, the electrode fabrication was complete, as shown in (g). (h) A 300 nm ${\mathrm{SiO}}_{2}$ layer was sputtered as a protective coating to prevent oxidation of the electrode. (i) After completing photolithography, three edges of the cantilever beam were etched, stopping at the ${\mathrm{SiO}}_{2}$ layer. (j) At this point, the front processing of the sensor was completed, and positive photoresist was applied for protection. (k) The back cavity of the cantilever beam was etched, similarly stopping at the ${\mathrm{SiO}}_{2}$ layer. (l) Vapor release was performed to etch away the intermediate ${\mathrm{SiO}}_{2}$, completing the wafer fabrication process.

## 4. Performance Analysis and System Testing

In Section 2, we described the design and parameter optimization of the SAW micro-pressure sensor. The obtained model parameters were imported into the Comsol finite element simulation software to compute its $S_{21}$ parameters through simulation. Figure 8 presents the simulation results, showing a resonant frequency of 438 MHz, a loss of −8.6 dB, and a 3 dB bandwidth of 0.36 MHz. The calculation of the Q factor satisfied Equation (Equation 9).(9)$$Q=\frac{f}{\Delta f_{3\mathrm{dB}}}$$

*f* is the resonant frequency, $\Delta f_{3\mathrm{dB}}$ is the 3 dB bandwidth, and the calculated *Q* value was 442.

**Figure 8 sensors-25-01873-f008:**
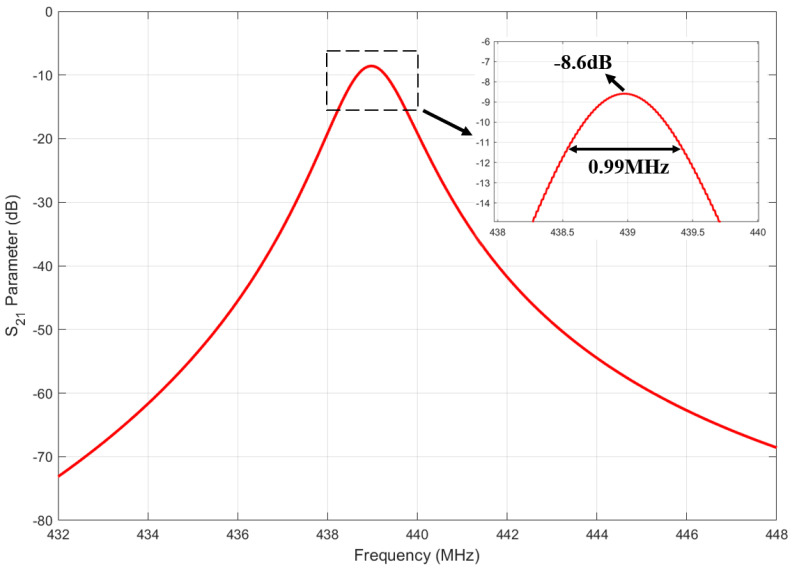
Static $S_{21}$ parameter results of SAW micro-pressure sensor obtained through Comsol simulation.

To verify the performance enhancement of the micro-pressure sensor with the introduction of a cantilever beam structure, the cantilever beam was fixed at both ends, and the inertial structure simulation was omitted. Figure 9a compares the performance of the SAW sensors with and without the cantilever beam structure. The sensitivity of the micro-pressure sensor with the cantilever structure was 25 kHz/Pa, while the sensitivity of the sensor without the cantilever beam structure was 500 Hz/Pa. The comparison indicates that the introduction of the cantilever beam structure enhanced the sensor’s sensitivity by a factor of 50. Under the same cantilever beam structure conditions, we utilized Lamb’s dispersion to find the wave speed of the fundamental $S_{0}$ mode and designed the corresponding dimensions for the IDT simulation. Figure 9b shows the performance comparison between the $A_{2}$ mode Lamb wave and the $S_{0}$ mode Lamb wave, revealing that the detection sensitivity of the $A_{2}$ mode Lamb wave was improved by 9 kHz/Pa compared to the $S_{0}$ mode Lamb wave.

The size of a single sensor was 7000 × 1000 μm, and it needed to be fixed onto a PCB, as shown in Figure 10a. The left end of the PCB had five holes for soldering SMA RF connectors, facilitating the connection of the device to a vector network analyzer (AV36580A). Figure 10b shows a physical photograph of the sensor assembly, where eight sensors were integrated onto a single PCB due to manufacturing constraints. In contrast, Figure 10c provides a detailed schematic of an individual SAW sensor. Figure 10d illustrates the design of the detection system, where a pressure controller was positioned above the device to provide varying levels of pressure. The resonator was excited by connecting it to a vector network analyzer that measured the real-time resonant frequency. To ensure the stability of the system and the accuracy of the test, the system used a wired connection.

Figure 11 compares the $S_{21}$ parameter amplitude under stable-state (black) and manually induced pressure-loaded conditions (blue/red). Due to laboratory equipment constraints, precise micro-pressure application devices were unavailable. Instead, pressure stimulation was implemented through a manual tapping method, yielding two discrete tapping intensities. Although this approach precluded exact pressure quantification, it reliably produced progressive resonant frequency shifts corresponding to the different mechanical inputs. The measurement system comprised a benchtop vector network analyzer (AV36580A) and a hand-operated mechanical excitation setup, with all tests conducted under standard laboratory conditions. These results serve as a proof-of-concept validation, with metrological calibration and quantitative analysis to be addressed in future studies using dedicated instrumentation.

## 5. Conclusions

This study presented the design and analysis of an $A_{2}$-mode Lamb-wave SAW sensor utilizing aluminum nitride (AlN) piezoelectric material and a SOI substrate. The IDT dimensions were systematically determined and a cantilever beam structure was incorporated as an inertial element, with its optimal dimensions calculated to enhance performance. The simulation results revealed that the inclusion of the cantilever beam increased the sensitivity to 25 kHz/Pa, representing a 50-fold enhancement compared to configurations lacking the cantilever structure. Additionally, leveraging the $A_{2}$-mode Lamb wave improved sensitivity by 1.5 times compared to the fundamental mode. The influence of various reflective grating designs on acoustic wave loss was analyzed, leading to an optimized configuration for minimal loss. A MEMS-compatible fabrication process was employed to realize a SAW micro-pressure sensor based on a multilayer structure comprising ${\mathrm{SiO}}_{2}$, AlN, and SOI. Furthermore, a micro-pressure testing system was developed to evaluate the sensor, demonstrating that varying micro-pressures induced distinct shifts in the resonant frequency.

## Figures and Tables

**Figure 1 sensors-25-01873-f001:**
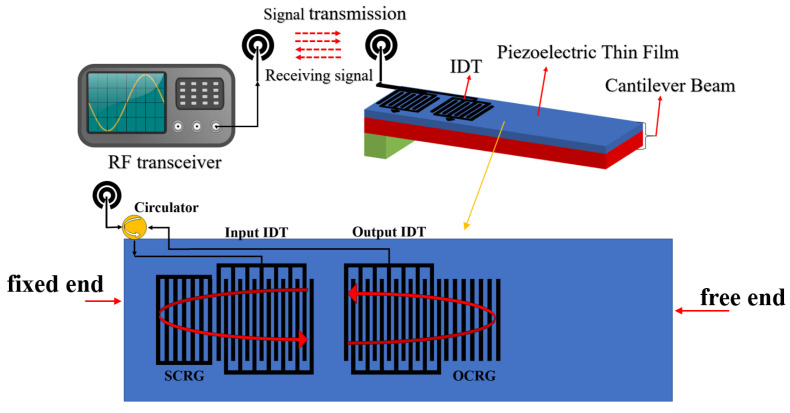
Schematic diagram of SAW micro-pressure sensor system.

**Figure 2 sensors-25-01873-f002:**
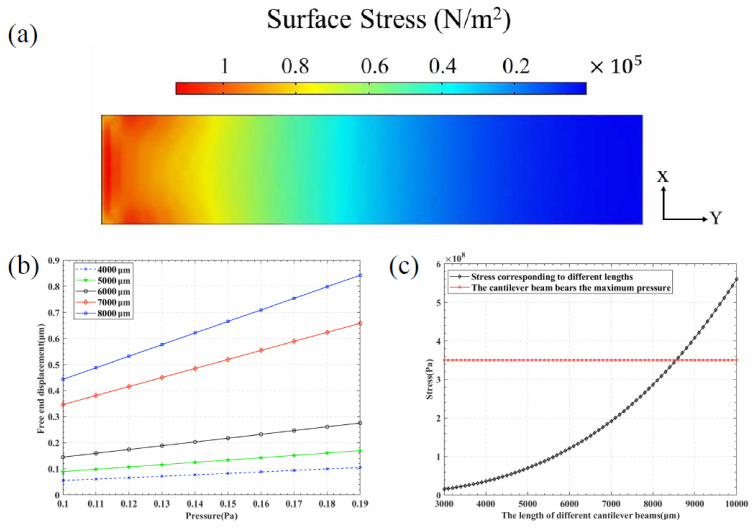
Simulation study of cantilever beam. (**a**) Stress diagram of two-dimensional cantilever beam. (**b**) Cantilever beams of different lengths experience free-end displacement under the same stress. (**c**) The maximum length of a cantilever beam without external force.

**Figure 3 sensors-25-01873-f003:**
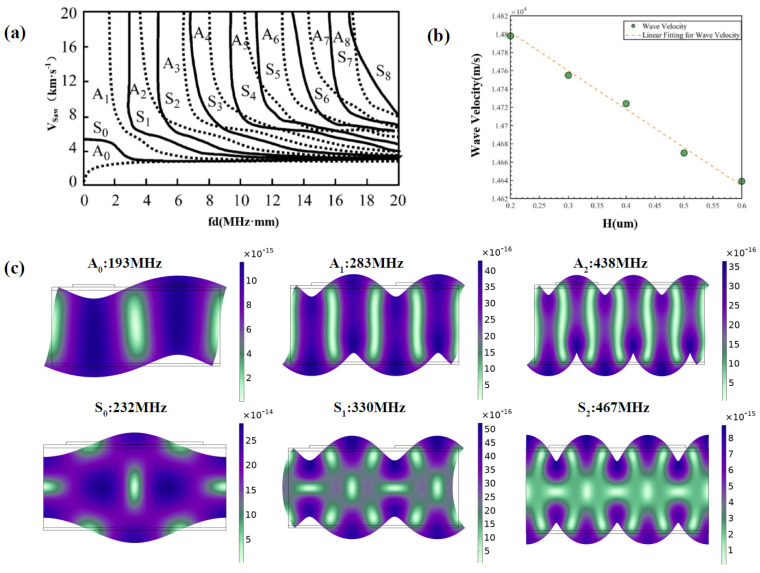
IDT simulation exploration. (**a**) Dispersion diagram of Lamb waves in AIL. (**b**) The influence of different electrode heights on Lamb wave velocity and linear fitting. (**c**) Different modal shapes of Lamb in AlN and Si.

**Figure 4 sensors-25-01873-f004:**
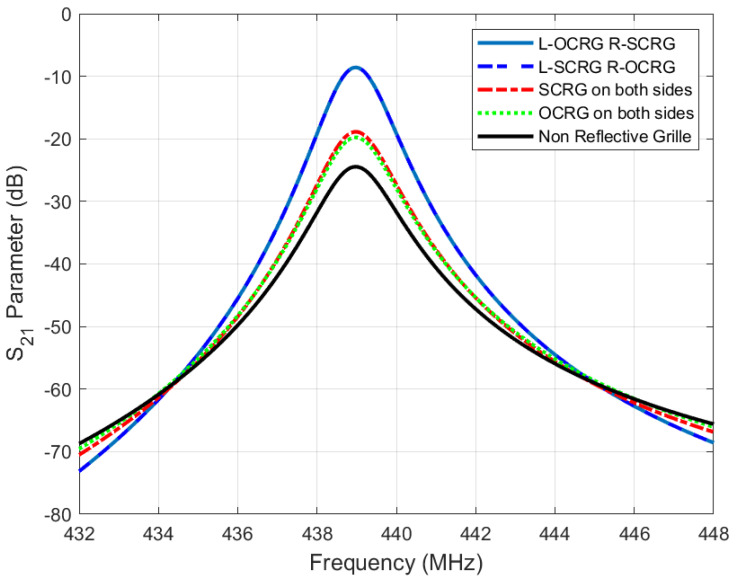
Simulation verification of different types of reflective grating effects.

**Figure 5 sensors-25-01873-f005:**
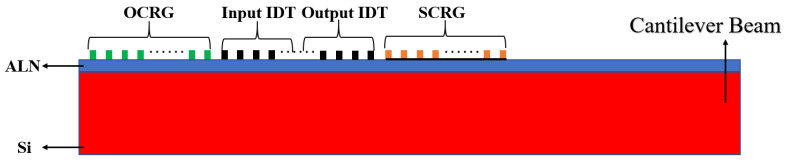
Simulation model completed by parameter optimization.

**Figure 6 sensors-25-01873-f006:**
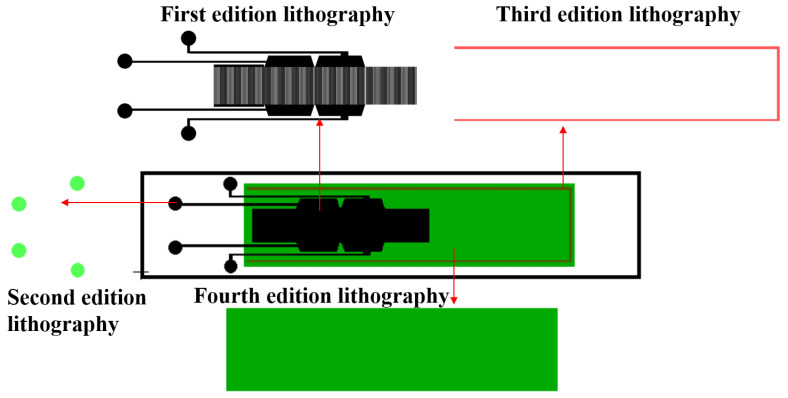
Four version lithography layout for MEMS compatible processes.

**Figure 7 sensors-25-01873-f007:**
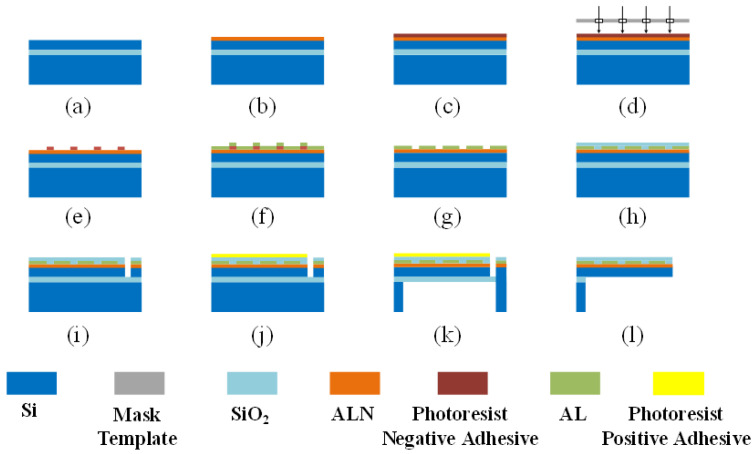
SAW micro-pressure sensor MEMS-compatible manufacturing process. (**a**) SOI surface cleaned. (**b**) AlN film sputtered. (**c**) Negative photoresist applied. (**d**) Mask exposed to UV. (**e**) Unexposed resist developed away. (**f**) Al sputtered; lift-off (NMP) performed. (**g**) Electrode completed. (**h**) ${\mathrm{SiO}}_{2}$ layer deposited. (**i**) Three cantilever edges etched (**j**) Protective positive resist applied. (**k**) Back cavity etched. (**l**) Cantilever beam freed.

**Figure 9 sensors-25-01873-f009:**
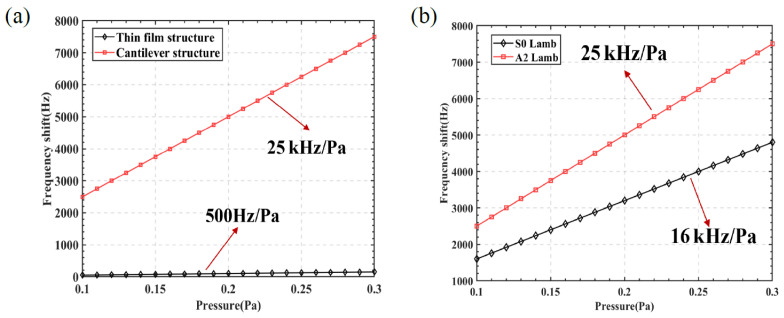
Performance comparison of SAW micro-pressure sensors. (**a**) Performance comparison of sensors with cantilever beam structure and without cantilever beam structure. (**b**) Performance comparison of different modal sensors.

**Figure 10 sensors-25-01873-f010:**
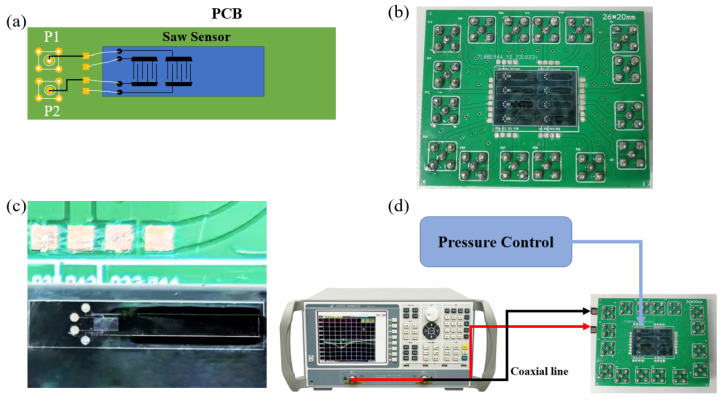
SAW micro-pressure sensor experiment. (**a**) Schematic diagram of SAW sensor. (**b**) Physical image of SAW sensor. (**c**) Detailed schematic of a single sensor. (**d**) Experimental testing system.

**Figure 11 sensors-25-01873-f011:**
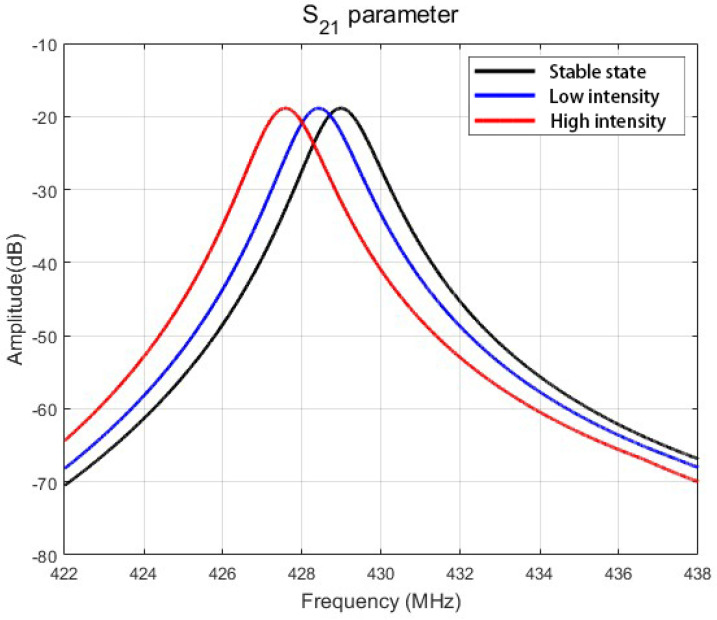
Physical performance test results of SAW micro-pressure sensor.

**Table 1 sensors-25-01873-t001:** Comparison of the main parameters related to pressure sensors based on surface acoustic wave (SAW) technology.

Author	Sensitivity	Wave Type	Substrate	Structure	Wireless
Quintero et al. [10]	8.3 kHz/bar	Rayleigh Wave	ST-cut Quartz	Circular Diaphragm	Yes
Rodríguez-Madrid et al. [11]	330 kHz/bar	Rayleigh Wave	AIN on CVD Nanocrystaline	Free-standing	No
Grousset et al. [12]	25.8 kHz/bar	Rayleigh Wave	AT-Cut Quartz (YXl)/37°	Membrane	No
Hu et al. [13]	589 ppm/Mpa	Rayleigh Wave	128° Y-X LiNbO_3_	Chamber Structure	Yes

**Table 2 sensors-25-01873-t002:** Parameters of common piezoelectric substrate materials.

Material Name	SAW Speed (m/s)	Electromechanical Coupling Coefficient (%)	Propagation Loss (dB/cm)
${\mathrm{SiO}}_{2}$	3301	1.5	0.35 (1 GHz)
AlN	5000	6.5	0.31 (1 GHz)
${\mathrm{LiNbO}}_{3}$	3506	3.5	0.26 (1 GHz)

**Table 3 sensors-25-01873-t003:** Parameters of the designed and simulated SAW micro-pressure sensor for verification.

Geometric Variables	Parameter
IDT width	8.5 μm
IDT interval	8.5 μm
Width of OSR and SRC	8.5 μm
Interval of OSR and SRC	8.5 μm
Interval between IDT and OSR	4.25 μm
Interval between IDT and SRC	8.5 μm
Electrode heights for IDT, OSR, and SRC	200 nm
Thickness of AlN	600 nm
Thickness of Si	15 μm

## Data Availability

The data supporting this research article are available upon request to the corresponding authors.

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
