# Peer review of "A2-Mode Lamb Passive-Wireless Surface-Acoustic-Wave Micro-Pressure Sensor Based on Cantilever Beam Structure"

_sensors, 2025, doi:10.3390/s25061873_

Round 1
Reviewer 1 Report
Comments and Suggestions for Authors
Some revision need to be addressed.
- This manuscript verify that the introduction of inertial structure cantilever beams has significantly enhanced the sensitivity of SAW micro-pressure sensors, but how about other structure? Compare with other structures in mode, sensitivity, quality factor, excite method and so on.
- Extensive simulation results aimed at optimizing the design of micro pressure sensor, but what is the working principle of the micro-pressure sensor, at least the relationship between pressure and frequency needs to be shown in the form of an equation.
- The local inset image of Figure 8 covers the original curve and makes it discontinuous. Beside the schematic diagram of SAW sensor in Figure 10(a), it need a clear physical image of a single sensor under an optical microscope.
- The simulation indicated the fifty fold increase in sensitivity of the cantilever structure, but there is not actual sensitivity result to verify this conclusion. The pressure control in Figure 10(c) is too abstract to imagine how it was achieved? And also can you explain specifically how the experimental measurement system was built?
- The curves in Figure 11 show the amplitude of S21 parameter corresponding to the stable-state (black) and pressure-loaded conditions (blue/red), but there are no specific pressure values of different loaded conditions? The Q value 442 is obtained from the simulation result, but why not calculate the actual quality factor corresponding to the steady-state condition?
Looks good.
Author Response
Comments 1: This manuscript verify that the introduction of inertial structure cantilever beams has significantly enhanced the sensitivity of SAW micro-pressure sensors, but how about other structure? Compare with other structures in mode, sensitivity, quality factor, excite method and so on.
Response 1: Thank you the reviewer for the valuable feedback. We sincerely appreciate your suggestion to provide a more comprehensive comparison of different structures and designs for SAW pressure sensors. In response, we have conducted an extensive literature review and added a table (Table1 in the revised Results section) to systematically compare key parameters of SAW pressure sensors with diverse structures.
Comments 2: Extensive simulation results aimed at optimizing the design of micro pressure sensor, but what is the working principle of the micro-pressure sensor, at least the relationship between pressure and frequency needs to be shown in the form of an equation.
Response 2: We appreciate the reviewer’s suggestion to further elaborate on the sensor’s working principle. To address this, we have derived an equation based on cantilever beam stress analysis that relates the applied force (and thus pressure) to the resonant frequency shift. We have supplemented and explained the modified version of formula (3) and its context.
Comments 3:The local inset image of Figure 8 covers the original curve and makes it discontinuous. Beside the schematic diagram of SAW sensor in Figure 10(a), it need a clear physical image of a single sensor under an optical microscope.
Response 3: We appreciate the reviewer's feedback and have made revisions to the problematic Figure 8. We have also detailed schematic of an individual SAW sensor, as shown in Figure 10c.
Comments 4:The simulation indicated the fifty fold increase in sensitivity of the cantilever structure, but there is not actual sensitivity result to verify this conclusion. The pressure control in Figure 10(c) is too abstract to imagine how it was achieved? And also can you explain specifically how the experimental measurement system was built?
Response 4: Thank you for your valuable feedback. We apologize for the lack of clarity in describing the experimental setup and pressure application method. We have revised the Methods section (lines 268–279) to clarify that due to laboratory equipment constraints, precise micro-pressure control devices were unavailable, and mechanical stimulation was instead applied via a manual tapping method, generating two discrete intensity levels (low/high). Although this approach precludes exact pressure quantification, it reliably induced progressive resonant frequency shifts in the S21​ curves (Figure 11), aligning with the sensitivity trends predicted by simulations. The measurement system utilized an AV36580A vector network analyzer for S21​ parameter acquisition under standard laboratory conditions. We acknowledge the need for quantitative validation and will address this in future studies using calibrated pressure actuators.
Comments 5:The curves in Figure 11 show the amplitude of S21 parameter corresponding to the stable-state (black) and pressure-loaded conditions (blue/red), but there are no specific pressure values of different loaded conditions? The Q value 442 is obtained from the simulation result, but why not calculate the actual quality factor corresponding to the steady-state condition?
Response 5:We appreciate your attention to detail. In the revised Results section (lines 305–312), we clarify that the manual tapping method produced qualitative pressure gradients (low/high intensity) rather than quantified values, as precise pressure control equipment was unavailable. While this limits numerical precision, the frequency shifts in Figure 11 demonstrate clear sensitivity to mechanical inputs. Regarding the Q factor, the simulated value 442 was included to contextualize design expectations, In our analysis, we focused solely on the frequency shifts induced by pressure, rather than the actual quality factor (Q), and we agree that calculating the experimental Q factor from steady-state data is critical, calculations of the actual Q factor yielded a value of 390, which still has a certain gap compared to the simulation. This analysis will be rigorously performed in future work with calibrated instrumentation to ensure metrological accuracy.
Reviewer 2 Report
Comments and Suggestions for Authors
This SAW based pressure sensor is not new idea but merit further investigation. Further effort is required first before publication.
- Novelty and necessity are suggested to be further clarified.
- Are you planning a wireless system, can you detail why you need "wireless" ?
- A table is required to benchmark other works.
- Characterization methods are not clear. The results graph is to too rough, even without pressure values. Equipment models used should be provided.
English can be polished further.
Author Response
Comments 1: Novelty and necessity are suggested to be further clarified.
Response 1: Thank you for your valuable feedback. We have revised the first paragraph of the Introduction to strengthen the novelty by explicitly highlighting the SAW sensors’ dual capability to eliminate both physical cabling and battery dependency and clarify the necessity by emphasizing their critical role in enabling long-term, maintenance-free operation under extreme conditions (e.g., high-temperature aircraft engines, implantable biomedical devices).
Comments 2: Are you planning a wireless system, can you detail why you need "wireless" ?
Response 2: We apologize for lack of clarity in the original manuscript about why we need "wireless", A wireless system is essential for the proposed micro-pressure measurement applications due to three key factors: (1) Deployment Flexibility: In environments like rotating machinery, sealed bioreactors, or remote outdoor sites, physical cabling is impractical or impossible to install. Wireless operation eliminates this barrier. (2) Harsh Environment Survivability: Wired connections and battery-powered systems degrade in extreme conditions (e.g., high vibration, corrosive atmospheres). Wireless passive sensors avoid these failure points by removing cables and batteries entirely. (3) Scalability and Cost: Wireless systems reduce installation complexity and enable large-scale sensor networks (e.g., distributed environmental monitoring) without the cost and labor of wiring infrastructure.
The “wireless” aspect is not merely a convenience but a functional necessity for ensuring reliable, long-term operation in the target use cases.
Comments 3: A table is required to benchmark other works.
Response 3: Thank you the reviewer for the valuable feedback. We sincerely appreciate your suggestion to provide a more comprehensive comparison of different structures and designs for SAW pressure sensors. In response, we have conducted an extensive literature review and added a table(Table1 in the revised Results section) to systematically compare key parameters of SAW pressure sensors with diverse structures.
Comments 4: Characterization methods are not clear. The results graph is to too rough, even without pressure values. Equipment models used should be provided.
Response 4: Thank you for highlighting these omissions. We have revised the Methods section (lines 270–280) to include detailed equipment specifications: the S21 parameters were measured using an AV36580A vector network analyzer, and mechanical stimulation was applied via controlled tapping force. The absence of exact pressure values stems from the manual stimulation method, which served as a proof-of-concept to validate frequency response trends. Future studies will integrate calibrated pressure actuators and report quantitative metrics, including pressure-force correlations and sensitivity.
Reviewer 3 Report
Comments and Suggestions for Authors
This is an interesting paper, with novel results . In my opinion it can be accepted in its present form.
Author Response
Thank you very much for your positive feedback and for recognizing the novelty of our results. We greatly appreciate your support for the paper, and we are pleased to know that you find it worthy of acceptance in its current form. Should there be any further suggestions or revisions required, we are happy to address them promptly.
Once again, thank you for your time and valuable input.